# HPLC-DAD-qTOF Compositional Analysis of the Phenolic Compounds Present in Crude Tomato Protein Extracts Derived from Food Processing

**DOI:** 10.3390/molecules26216403

**Published:** 2021-10-23

**Authors:** Ana Miklavčič Višnjevec, Paul W. Baker, Kelly Peeters, Matthew Schwarzkopf, Dominik Krienke, Adam Charlton

**Affiliations:** 1Faculty of Mathematics, Natural Sciences and Information Technologies, University of Primorska, Glagoljaška 8, SI-6000 Koper, Slovenia; ana.miklavcic@famnit.upr.si; 2The Biocomposites Centre, Bangor University, Bangor LL57 2DG, UK; paul.baker@bangor.ac.uk; 3InnoRenew CoE, Livade 6, SI-6310 Izola, Slovenia; kelly.peeters@innorenew.eu (K.P.); matthew.schwarzkopf@innorenew.eu (M.S.); 4Andrej Marušič Institute, University of Primorska, Muzejski trg 2, SI-6000 Koper, Slovenia; 5GEA Westfalia Separator Group GmbH, Werner-Habig-Straße 1, 59302 Oelde, Germany; Dominik.Krienke@gea.com

**Keywords:** phenolic compounds, HPLC-DAD-qTOF, *Solanum lycopersicum* L., protein extracts, agricultural residues

## Abstract

The conversion of raw fruits and vegetables, including tomatoes into processed food products creates side streams of residues that can place a burden on the environment. However, these processed residues are still rich in bioactive compounds and in an effort to valorize these materials in tomato by-product streams, the main aim of this study is to extract proteins and identify the main phenolic compounds present in tomato pomace (TP), peel and skins (TPS) by HPLC-DAD-ESI-QTOF. Forty different phenolic compounds were identified in the different tomato extracts, encompassing different groups of phenolic compounds, including derivatives of simple phenolic acid derivatives, hydroxycinnamoylquinic acid, flavones, flavonones, flavonol, and dihydrochalcone. In the crude protein extract (TPE) derived from tomatoes, most of these compounds were still present, confirming that valuable phenolic compounds were not degraded during food processing of these co-product streams. Moreover, phenolic compounds present in the tomato protein crude extract could provide a valuable contribution to the required daily intake of phenolics that are usually supplied by consuming fresh vegetables and fruits.

## 1. Introduction

The development and optimization of technologies for the recovery of bioactive compounds in food waste and subsequent valorization of these compounds in a range of industrial applications, including functional food ingredients, supplements or nutraceutical formulations is becoming an important solution to this challenge [1]. Among the biologically active compounds present in agricultural production residues are an important group of functional phenolic compounds. These are secondary metabolites which act to provide plant defense and protective mechanisms [2]. They have shown to have anti-inflammatory, antimicrobial, and antioxidant effects and they can have a protective role against various chronic degenerative and cardiovascular diseases, and cancer [3,4,5,6]. However, the intake, metabolism, and physiological effects all dietary antioxidants, including their interaction with other components in food must be taken into account when evaluating their health benefits [7]. For example, Goñi et al. [7] showed that the dietary intake of polymeric polyphenols by the elderly is predominantly associated with fiber matrix in foods, which potentially promotes improved gastrointestinal health. Certain proteins and peptides also exhibit antioxidant properties [8] and can contribute to antioxidant effects of phenolic compounds derived from plant protein extracts [9].

Synthetic phenolic antioxidants are widely used in the food industry because they can effectively extend the preservation time of oily food items [10]. The proposed maximum limits for synthetic phenolic antioxidants such as butylated hydroxyanisole (BHA), butylated hydroxytoluene (BHT), and tertiary butylhydroquinone (TBHQ) indicate that the acceptable daily intake might be approached or exceeded in certain countries [11]. Therefore, with the increasing use and their inevitable release into the environment, these groups of synthetic phenolic antioxidants have the potential to increase risks linked to the environment and human health [10]. Due to consumers demand for non-synthetic additives and in the search for environmentally and economically conscious choices, the use of by-products as a source of food additives is one of the most relevant potential solutions [12]. Moreover, agricultural production residues linked to food processing, including seeds and peels, often contain the highest levels of phenolic compounds [3,13]. Therefore, natural antioxidants extracted from different agricultural residues could be utilized in food processing applications including cooked meats where lipids, particularly their phospholipids, are susceptible to autoxidation [14]. In fact, antioxidants such as phenolic compounds are only naturally present in smoked meats and not in other meat products [15], which requires the addition of synthetic or naturally present antioxidants [16] to improve shelf-life and stability.

Tomato (*Solanum lycopersicum* L.) is one of the most widespread fruits in the world and the bioactive components present can be broadly categorized as either carotenoids or phenolic compounds. In tomatoes, one of the major bioactive molecules present is the carotenoid, lycopene [17]. Phenolic compounds are also present in tomatoes in lower concentrations than carotenoids [17] and most of these belong to hydroxycinnamic acids and flavonoids such as flavanones, flavonols, and anthocyanidins [18]. Flavonols are the predominant group of flavonoids found in tomato with quercetin, kaempferol, and myricetin the main compounds, with naringenin present in higher concentrations in some varieties. Anthocyanidins such as cyanidin, pelargonidin, and delphinidin are present in lower concentrations, along with hydroxycinnamic acids derivatives including chlorogenic, caffeic, ferulic, 4-*O*-caffeolyquinic, and p-coumaric acids [18,19]. 

In one study the main phenolic compounds present in the tomato peels were identified as quercetin and kempferol [20], however, another report [21], determined that rutin and naringenin were the main phenolic compounds l, while rutin, chlorogenic acid, and quercetin derivatives were present in minor quantities. However, the compositional profile and concentration of phenolic compounds in tomatoes are strongly dependent on the tomato variety [17,19] and also significantly influenced by maturity, harvesting time, and production method [18,22].

The tomato pulp and seeds are the main discarded fractions produced from tomato processing and are often used as a source of animal feed [23]. They could be also a good source for the production of protein concentrates [24] for use as food ingredients [9]. Seventeen different amino acids were identified in tomato, including essential amino acids which comprised ~40% of the total protein that could be extracted [24]. There is therefore considerable potential for use of the protein and phenolic compounds components present in tomato processing residues for different applications in the food ingredients and supplements sectors.

In this study, an analysis of the bioactive compounds present in tomato processing residues was conducted, with a focus on the phenolics present in each of the fractionated streams including the pomace (TP) peels and skin, (TPS) and the protein seed extract (TPE). As far as we are aware this is the first study reporting the phenolic composition in different tomato fractions and the derived protein extract.

## 2. Results and Discussion

### 2.1. Identification of Phenolic Compounds in Tomato Samples

Valverdu-Queralt et al. [25] categorized the phenolic compounds found in tomato-based products into derivatives of the following: (1) simple phenolic acids; (2) hydroxycinnamoylquinic acids; (3) flavones; (4) flavonones; (5) flavonols, and (6) dihydrochalcones. The additional phenolic compounds identified in the different tomato processing fractions analyzed during this study are discussed in relation to these different groups.

The phenolic compounds identified in different tomato samples are presented in the Table 1. The main peaks in the UV chromatogram at 280 nm of selected samples were annotated (Appendix A).

#### 2.1.1. Simple Phenolic Acid Derivatives

In the various tomato extracts, coumaric (1), protochatechuic (14), and caffeic (11) acids were identified, which exhibited typical fragmentation patterns, including a characteristic loss of carbon dioxide as previously reported [26]. Protocatechuic acid (14) was found in all three extracts at approximately the same retention time with the characteristic fragmentation pattern. Caffeic acid (11) was only observed in the TP and in the TPE, which was identified by its exact mass and expected fragmentation pattern. Two possible isomers of coumaric acid (m/z 153) were identified in the tomato extracts with retention times of 3.4 and 7.2. In the last protein sample analyzed, the first coumaric acid isomer was not observed. The relative retention times for caffeic and protocatechuic acids were similar to those previously reported [25], where both compounds were distinguished based on exact mass and fragmentation pattern (Table 1).

Simple phenolic acids glucosides were found in the TP and the TPE, including the glucosides of caffeic (3, 6, 16), homovanilic (4), coumaric (10), and ferulic acids (12, 15), as previously reported in tomato [27]. The presence of vanillic acid glucoside in whole tomatoes has been reported previously [27], although in our study this was not confirmed by fragmentation pattern due to its low concentration. Different isomers of caffeic and ferulic acid glucoside were determined based on their expected masses, fragmentation patterns, and elution times. Caffeic acid and ferulic acid glucoside (3, 6, 16, 12, 15) were present in the TP, but not in TPS (Table 1).

#### 2.1.2. Hydroxycinnamoylquinic Acid Derivatives

Chlorogenic acid (RT = 6.1, 5) and its isomers including 5-*O*-caffeoylquinic acid (RT = 5.3, 2), criptochlorogenic acid (6.3, 8) with molar mass 354 g/mol, were identified in the tomato extracts. While chlorogenic acid (5) was identified using an analytical standard, other isomers were determined based on their retention times and relative intensities of the associated fragments [28]. Chlorogenic acid (5) was determined in all three extracts and the presence of criptoclorogenic acid (8) was confirmed in TP and TPE, while 5-*O*-caffeoylquinic acid (2) was identified only in TP.

Only one isomer of feruloyl quinic acid (20) was identified during the study and that was found in TPS. The compound was identified according to prevalence of 191 fragment ion as previously reported [29]. In addition, coumaryl quinic acid (17) was also only present in the TP at the retention time 6.9, with a typical fragmentation pattern. Finally, dicaffeoylquinic acid isomers (26, 27, 30) were identified with m/z 515 with typical fragmentation patterns. The first two with retention times 7.9 and 8.0 (26, 27) were found in all three extracts, while the third was found only in the TPS (30).

#### 2.1.3. Flavone Derivatives

One of the main flavone derivatives present in tomatoes is apigenin (39) that has been shown to possess anti-inflammatory, antioxidant, and anticancer properties [30]. It is considered safe even at higher doses, and no toxicological issues with this compound have been reported [31]. However, at high doses it can trigger muscle relaxation and sedation [32]. Apigenin was found in the TPE (39) and confirmed by the standard (Table 1). Likewise, apigenin-7-*O*-glucoside (3) was identified based on a previous report [33] only in the TPE. Vicenin-2 or apigenin-6,8-di-C-glucopyranoside (13) was tentatively identified in the TPE, based on a previous report of this compound [25].

#### 2.1.4. Flavanone Derivatives

Naringenin (38) is one of the main flavonoids in TPS [27] and its pharmacological impacts on human health are well described in the literature, including its potential use in treating osteoporosis, cancer, and cardiovascular disease [34]. Therefore, it was not surprising that naringenin (38) was present in high levels in all three extracts analyzed in the current study. In the TP, an additional isomer of naringenin (40) was present at retention time of 10.7, with the exact molar mass and characteristic fragmentation pattern. Structurally similar compounds including eriodictyol (36) were identified through typical fragmentation patterns and found in both TP and TPE, but not in the TPS. Based on the reported fragmentation pattern, it is possible to distinguish between naringenin-*O*-glucoside (22, 33, 34) and naringenin-C-glucoside (21) [25], and the presence of the fragment with m/z 271 is possibly due to an *O* isomer of a hexoside or glucoside. It may also be possible that fragments m/z 343 and 313 might be a consequence of characteristic losses of m/z 90 and m/z 120, due to cross-ring cleavages in hexose unit. In our study, one C isomer (21) and three *O* isomers (22, 33, 34) were found of naringenin-glucoside (Table 1). All four isomers were present in the TPE, while just a C isomer was present in the TPS and one *O* isomer was present in the TP. 

Similarly to the naringenin-C-glucoside (21), naringenin-C-diglycoside (25) (m/z 595) was tentatively identified due to the loss of m/z 90 and 120 in both the TP and TPS fractions (Table 1). In the protein extract, eriodictyol-*O*-glucoside (32) was tentatively identified at retention time 8.4, with a corresponding fragmentation pattern as previously reported by Vallverdu-Queralt et al. [25]. Both the deprotonated molecule (m/z 449) and hexoside moiety (m/z 287) possessed similar m/z of 449 at a retention time of 8.4 (32). In addition, at retention times 7.7 and 8.7, a hexoside moiety was detected as the main fragment at m/z 449. Consequently, two additional isomers were tentatively identified and linked to eriodictyol-*O*-hexoside (24, 25). 

#### 2.1.5. Flavonol Derivatives

A bitter-tasting flavonol glycoside, kampferol-3-*O*-rutinoside (31) was identified in the TP and TPE, with a retention time of 8.5, and based on accurate mass determination and a typical fragmentation pattern, with a deprotonated ion (m/z 593) and the loss of a rutinoside unit (308 Da) with m/z 285 (Table 1, peak number 31). Similar compounds, including kampferol-3-*O*-rutinoside (31), quercetin-3-*O*-rutinoside (rutin, 23), a well-known compound and widely distributed in edible plants [35], were also identified and assigned on the basis of accurate mass determination, similar retention times, and typical fragmentation pattern reported previously in olives [33]. In addition, quercetin (m/z 301, 37) was identified in both TP and TPE, with a fragmentation pattern following a retro Diels-Alder (RDA) process previously described [36]. In addition, rutin-*O*-hexoside (7) was tentatively identified through the following fragments: 771, 609, and 300. This compound was found in the TP and TPE. The presence of rutin-*O*-pentoside (19) was also tentatively identified in all three extracts with m/z 741 and the main fragments 741 and 300 (Table 1). Quercetin-3-galactoside (hyperoside, 28) was identified in TP and TPE according to the exact mass and fragmentation pattern reported in the literature [37].

### 2.2. Phenolic Compound Compositional Profile of the Three Tomato Processing Fractions

The main peaks that were annotated in the UV chromatograms of the raw material (pomace) and the final product (TPE) were compared. In addition, the same compounds found in these two types of extracts were compared with the TPS. The heatmap for semi-quantitative comparison of the main phenolic compounds identified is shown in the Figure 1.

Caffeic acid-*O*-hexoside, homovanillic acid glucoside, chlorogenic acid, caffeic acid-*O*-hexoside, cryptochlorogenic acid, caffeic acid, coumaric acid, and rutin-*O*-pentoside were present in much higher concentrations in the TP containing seeds compared to the TPE. Rutin-*O*-hexoside, ferulic glucoside, rutin, and naringenin-*O*-glucoside were also present in the same quantities in both extracts, while protocatechuic acid, eriodictyol, and quercetin were present at higher amounts in the TPE fraction compared to TP. Most of the compounds present in the TPE were not found in the TPS. Chlorogenic, protocatechuic, and coumaric acids along with rutin-*O*-pentoside were present in much lower concentrations, while rutin was present at approximately the same range and naringenin at a higher concentration compared to TPE. Overall, the different phenolic compounds found in the TPE are also probably due to the removal of those compounds from the original pomace, peels, and skins during processing and washing of the final product.

It is important to note that overall, in the TPE, most phenolic compounds were still present after processing. Together with the protein (Table 2), the presence of key phenolic compounds determined in the TPE could be a good source of natural antioxidants suitable for use by the food ingredients and supplements sectors TPE contains 220 ± 15 µg/g dry weight of total determined phenolic compounds. This concentration is in the same range as the levels of total phenolic compounds determined by HPLC-DAD-MS/MS in dry tomato [19]. A previous report [9] evaluated protein concentrates produced from *Amaranthus mantegazzianus**,* an annual flowering plant, and reported the different phenolic compound content and antioxidant activity in these concentrates. It was concluded that the high antioxidant activity in the water extract of the whey fraction could be related to high protein content of this extract, while the high reported antioxidant activity in methanolic whey extracts could be linked to the high phenolic compounds content. The most important conclusion of this study was that the protein extracts evaluated could be suitable to use as additives to enhance both the nutritional and health-related aspects of various food products [9]. This is in accordance with our observations, however, it would be advantageous to see additional studies involving protein-phenolic compound interactions in tomato protein extracts and the potential impact on both the digestibility and functionality of these materials in relation to their reported anti-viral, anti-cancer, and anti-inflammatory properties [14,38]. Further studies are required to confirm these observations. In addition, TPE could provide a valuable contribution to the required daily intake of phenolic compounds that are usually supplied by consuming vegetables and fruit, especially considering the daily consumption of fruit and vegetables is usually far below the recommended values—for example, according to a health Survey for England, only 28% of adults were found to be eating fruits and vegetables according to the recommended five portions per day [39]. 

## 3. Materials and Methods

### 3.1. Sample Description

#### Tomato Samples

Tomato pomace (peel, outer skins, and seeds) originated from Agrofusion, Ukraine. Tomato pomace (50 kg) is the remnants of salad tomatoes after collecting the juice by heating at 80 °C, squeeze pressed, and then shipped frozen. The pomace was separated into pulp and peel and into seeds, followed by protein extraction from the seeds.

### 3.2. Separation of Tomato Pomace into Different Fractions

All equipment were washed with 1% detergent and then rinsed with sanitizer. The contents were heated to 50 °C for 15 min in water (200 L) with slow stirring until completely defrosted. These seeds no longer possessed an outer gel layer and the peel and skins were skimmed from the surface using a sieve with 1 mm holes with slow rotation. The contents of the hot pan were repeatedly drained into 30 L buckets and any peel appearing on the surface was skimmed off with the sieve. The seeds were collected on a sieve with 0.5 mm holes and the seeds were weighed. Moisture analysis was performed at 105 °C until the loss of moisture was less than 20 mg per min. The dry weights were calculated based on the moisture contents to reveal that the pomace was composed of 70.4% seeds and 29.6% of pulp and peel. 

### 3.3. Extraction of Crude Protein from Tomato Seeds

The tomato seeds (698 g wet weight equivalent to 200 g in dry weight) were immersed in deionized water (4 L) and high shear mixed (Silverson mixer) using the workhead with the largest holes (general purpose disintegrating head) for 5 min at 7000 rpm. The workhead was changed to one with smaller holes (square hole high shear screen) and mixed for 5 min at 7000 rpm. Finally, the workhead was changed again with one with the smallest holes (emulsor screen) and mixed for 5 min at 7000 rpm. The suspension was sieved, and the seeds were re-extracted with deionized water (3 L) with high shear mixing at 7000 rpm for 5 min using the emulsion screen. This was repeated for a second time in order to recover three filtrates. Each of the filtrates were left to settle for 5 min and the top layer was decanted leaving behind the sedimented material that had passed through the sieve. To each of the filtrates, 4 M HCl was added to adjust from pH 6.64 ± 0.25 to pH 4.01 ± 0.05. The primary filtrates required more acid. The protein suspensions were cooled at 4 °C for 1 h and then the colloidal suspension was manually decanted ensuring that none of the sediment was discarded. The remaining sedimented protein was centrifuged at 3000 rpm for 15 min in the Beckman centrifuge and the supernatant was discarded. Some of the colloidal protein in suspension could not be precipitated even with high-speed centrifugation.

Before the analysis, excess water was removed from the remaining peels and skins by passing them through the juicing machine using the second largest filter and then the remaining material was weighed. Further water was removed from the material by squeeze pressing.

### 3.4. Protein Content

Crude protein was recovered after each successive extraction step, indicating in terms of protein yield compared with the original quantity recovered during the first extraction that one-third and one-tenth were recovered in the second and third extractions, respectively. Altogether this formed about two-thirds of the total tomato seed biomass. The weight of the seeds at the start indicated that 8 g was lost as soluble compounds or fats and oils when the weights of the remaining seed hulls and crude protein extracts were subtracted. The crude protein obtained after the first extraction step was slightly higher compared to the crude protein obtained during the later steps. However, the protein content of first extract determined by Kjeldahl analysis was higher at 31%. The protein content determined in the crude protein extracted that was performed using Kjeldahl analysis quantifies all the protein, both soluble and insoluble. In contrast, protein concentrations determined using the Bradford assay quantifies only protein containing an open structure that is accessible to the Bradford reagents. Therefore, the difference between both measurements could be attributed to protein that was less accessible to the Bradford reagents [40]. The protein content associated with the remaining seed hulls was lower than the protein content associated with any of the crude protein extracts. The combined protein yields from the remaining seed hulls and protein extracts revealed a difference of 60% compared with the protein content determined in the original seeds. Some of this protein may have been lost as soluble protein that could not be precipitated although considering that most of the biomass was recovered as remaining seed hulls and as protein extracts, this is unlikely to account for all of the protein difference. Therefore, it is possible that a significant proportion of protein remained with one of these components. 

### 3.5. Phenolic Compounds Determination by HPLC-DAD-qTOF

The extraction method for HPLC-DAD-qTOF analysis of phenolic compounds was adopted from Barros et al. [19]. Each sample was extracted with methanol: water (80:20 *v*/*v*) at ambient temperature, with agitation (150 rpm) for 1 h and then filtered through Whatman No. 4 paper. The residue was re-extracted twice with additional 25 mL portions of the same solvent. The combined extracts were evaporated at 35 °C under vacuum to remove the solvent. The crude extracts were diluted with 1 mL of methanol: water (80:20 *v*/*v*) and filtered through a 0.2 µm/PA (Nylon) filters before analysis using HPLC-ESI-qTOF.

The phenolic compounds were characterized using a high-pressure liquid chromatography system (HPLC, Agilent 317 1290 Infinity 2 HPLC modules, Santa Clara, CA, USA), interfaced with a electrospray ionization-quadrupole time-of-flight (ESI-qTOF) mass spectrometer (6530 Agilent Technologies, Santa Clara, CA, USA). An HPLC was equipped with a Poroshell 120 column (EC-C18; 2.7 µm; 3.0 × 150 mm, Agilent, Santa Clara, CA, USA) and an elution gradient of water/formic acid (99.05:0.5, *v*/*v*) (A) and acetonitrile/methanol (50:50, *v*/*v*) (B) was used for 20 min (flow rate: 0.5 mL min; injection volume: 1 uL, column temperature 50 °C) starting at 3.0% B increasing to 100.0% B in 15 min and maintained at this concentration for 5 min [41]. The separated compounds were first monitored using DAD (280 nm) and then MS scans were performed in the range m/z 40–1000, using the following conditions: capillary voltage, 2.5 kV; gas temperature 250 °C; drying gas 8 L/min; sheath gas temperature 375 °C; sheath gas flow 11 L/min (accuracy within ± 3 ppm). Automated MS/MS data-dependent acquisition was performed for ions detected in the full scan above 2000 counts with a cycle time of 0.5 s, using the collision energies: 10, 20, and 40 eV. The instrument was tuned in low mass range up to 1700 m/z and in extended dynamic range 2 GHz in negative mode. All data were processed using Qualitative Workflow B.08.00 and Qualitative Navigator B.080.00 software. The extracts were screened for the range of phenolic compounds previously reported in tomato and identified based on accurate mass and fragmentation pattern profile obtained from METLIN (Metabolite and Chemical Entity Database), standard solutions of a chlorogenic acid (Sigma-Aldrich, Merch KGaA, Darmstadt, Germany), and apigenin (Sigma-Aldrich, Merch KGaA, Darmstadt, Germany) or literature data [25,26,27,28,33,36]. In addition, the main identified phenolic compounds were quantified: caffeic acid-*O*-hexoside (3), homovanillic acid glucoside (4), chlorogenic acid (5), caffeic acid-*O*-hexoside 2 (6), rutin-*O*-hexoside (7), cryptochlorogenic acid (8), caffeic acid (11), ferulic acid glucoside 1 (12), vicenin-2 (13), protocatechuic acid (14), coumaric acid 2 (18), rutin-*O*-pentoside (19), rutin (23), naringenin-*O*-glucoside 3 (34), eriodictyol (36), quercetin (37) and naringenin 1 (38). Phenolic compounds were quantified using the response factor for chlorogenic acid (Sigma-Aldrich, Merck KGaA, Darmstadt, Germany). The calibration plots indicated good correlations between peak areas and commercial standard concentrations. LOQ was determined as the signal-to-noise ratio of 10:1 and amounts to 0.1 mg/kg dry weight sample. The standard deviation between duplicates was less than 7%.

## 4. Conclusions

In order to support the development of innovative extraction methodologies for functional secondary metabolites present in tomato processing residues, the phenolic compound compositional profile was determined for: tomato pomace, peel, and skins that was separated from the pomace; and crude tomato protein extract, using HPLC-DAD-ESI-QTOF. Forty different phenolic compounds were identified in these different tomato processing fractions, including derivatives of phenolic and hydroxycinnamoylquinic acids, flavones, flavonones, flavonols, and dihydrochalcones. In this preliminary study the most important finding was that most of these compounds were still present in the final protein extract and remained undegraded during processing of the tomato pomace. This crude protein could provide a valuable contribution to the required daily intake of phenolics that are usually supplied by consuming vegetables and fruits. Concentrating and boosting the levels of phenolics present through the use of food supplements and ingredients containing these compounds may help improve human health.

## Figures and Tables

**Figure 1 molecules-26-06403-f001:**
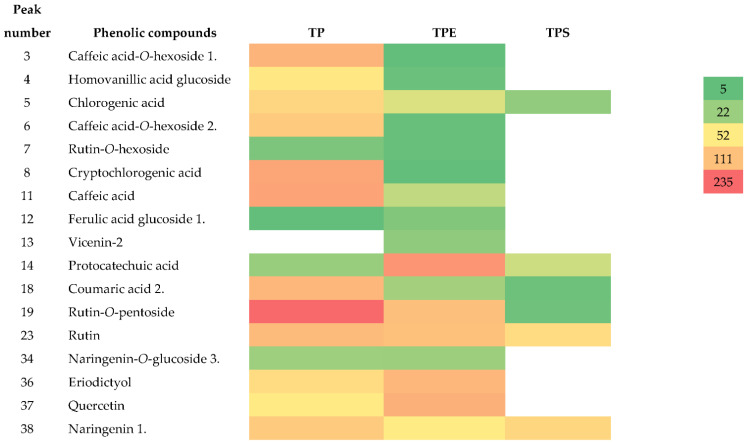
Heatmap representing the results of semi-quantitative comparison of individual phenolic compound identified in the tomato pomace containing seeds (TP), tomato protein (TPE), and tomato skins and peels (TPS). The relative abundances (relative intensity) detected by the mass spectrometer are shown with different colors. The deeper is the red color, the higher is the relative mass abundance of the phenolic compound; the deeper is the green color, the lower is the relative mass abundance of the phenolic compound. Calculations were performed based on areas of MS extracted ion chromatogram (EICs). The data normalization was performed including the correction for dilution during sample preparation.

**Table 1 molecules-26-06403-t001:** Phenolic compounds present in tomato extracts.

Peak Number	Compound	Extract *	RT	m/z [M]^−^	Fragments	Molecular Formula
1	Coumaric acid 1	1 2	3.4 2.3	163.0406	163.0361, 119.0484	C_9_H_8_O_3_
2	5-*O*-caffeoylchlorogenic acid	1	5.3	353.0890	191.0549, 179.036, 135.0435	C_16_H_18_O_9_
3	Caffeic acid-*O*-hexoside 1	1 3	5.5 5.1	341.0880	179.0343, 119.0300	C_15_H_18_O_9_
4	Homovanillic acid glucoside	1 3	5.9 5.9	343.1036	343.1941, 137.0625, 109.0597	C_15_H_20_O_9_
5	Chlorogenic acid	1 2 3	6.1 6.1 5.8	363.0880	191.0540	C_16_H_18_O_9_
6	Caffeic acid-*O*-hexoside 2	1 3	6.1 5.8	341.0879	179.0345, 135.0406	C_15_H_18_O_9_
7	Rutin-*O*-hexoside	1 3	6.2 6.0	771.2006	771.2017, 609.1429, 300.0212	C_33_H_40_O_21_
8	Cryptochlorogenic acid	1 3	6.3 6.0	353.0875	191.0576, 173.0459, 135.0433	C_16_H_18_O_9_
9	Naringenin-C-diglycoside	1 2	6.4 6.5	595.1675	505.1202, 475.1244, 385.0929, 355.0846	C_27_H_32_O_15_
10	Coumaric acid glucoside	1	6.4	325.0927	163.0396, 119.0510	C_15_H_18_O_8_
11	Caffeic acid	1 3	6.6 6.3	179.0358	135.0442, 179.0329	C_9_H_8_O_4_
12	Ferulic acid glucoside 1	1 3	6.7 5.7	355.1060	193.0489, 178.0345, 149.0512	C_16_H_20_O_9_
13	Vicenin-2	3	6.6	593.1517	473.0995, 353.0639	C_27_H_30_O_15_
14	Protocatechuic acid	1 2 3	6.7 6.5 6.1	153.0194	153.0188 109.029	C_7_H_6_O_4_
15	Ferulic acid Glucoside 2	1	6.7	356.1107	193.0489, 178.0345, 149.0512	C_16_H_20_O_9_
16	Caffeic acid-*O*-hexoside 3	1	6.8	341.0893	135.0386, 179.0339	C_15_H_18_O_9_
17	Coumaroylquinic acid	1	6.9	337.0948	191.0521, 163.0376	C_16_H_18_O_8_
18	Coumaric acid 2	1 2 3	7.2 7.6 7.3	163.0397	163.0416, 119.0502	C_9_H_8_O_3_
19	Rutin-*O*-pentoside	1 2 3	7.2 7.4 7.3	741.1891	300.0239, 741.1882	C_32_H_38_O_20_
20	Feruloylquinic acid	2	7.3	367.1042	191.0516	C_16_H_18_O_8_
21	Naringenin-C-glucoside	2 3	7.3 7.5	433.1168	433.1228, 343.0811,	C_21_H_22_O_10_
22	Naringenin-*O*-glucoside 1	3	7.6	433.1147	433.1203, 271.0590	C_21_H_22_O_10_
23	Rutin	1 2 3	7.7 7.9 7.8	609.1469	609.1444, 300.0347, 179.0006	C_27_H_30_O_16_
24	Eriodyctyol-*O*-glucoside 1	3	7.7	449.1097	287.0549	C_21_H_22_O_11_
25	Phloretin-C-diglycoside	2 3	7.8 7.9	597.1833	447, 387.1110, 357.0980, 417.1134	C_27_H_34_O_15_
26	Dicaffeoylquinic acid 1	1 2 3	7.9 8.1 7.9	515.1179	515.1241, 353.0863, 191.0566, 173.0416, 335.0748	C_25_H_24_O_12_
27	Dicaffeoylquinic acid 2	1 2 3	8.0 8.2 8.0	515.1206	515.1241, 353.0863, 191.0571, 173.0402, 335.0758	C_25_H_24_O_12_
28	Quercetin-3-galactoside	1 3	8.1 7.9	463.0917	271.0197, 255.0216, 300.0279, 243.0251	C_21_H_20_O_12_
29	Apigenin-7-*O*-glucoside	3	8.3	431.0996	431.1009, 269.0441	C_21_H_20_O_10_
30	Dicaffeoylquinic acid 3	2	8.4	515.1203	515.1054, 353.0859, 191.0570, 173.0429, 335.0619	C_25_H_24_O_12_
31	Kaempferol-3-*O*-rutinoside	1 3	8.5 8.4	593.1523	593.1492, 285.0380	C_27_H_30_O_15_
32	Eriodityol-*O*-glucoside 2	3	8.4	449.1111	287.0439, 449.1029	C_21_H_22_O_11_
33	Naringenin-*O*-glucoside 2	3	8.5	433.1146	433.2202, 271.0610	C_21_H_22_O_10_
34	Naringenin-*O*-glucoside 3	1 3	8.6 8.5	433.1168	433.2202, 271.0581	C_21_H_22_O_10_
35	Eriodityol-*O*-glucoside 3	3	8.7	449.1111	287.0598	C_21_H_22_O_11_
36	Eriodictyol	1 3	9.7 9.5	287.0563	151.0017, 135.0424	C_15_H_12_O_6_
37	Quercetin	1 3	10.0 9.9	301.0360	301.036, 150.9920	C_15_H_12_O_5_
38	Naringenin 1	1 2 3	10.2 10.5 10.4	271.0612	151.0029, 119.0504	C_15_H_10_O_7_
39	Apigenin	3	10.7	269.0475	269.4560	C_15_H_10_O_5_
40	Naringenin 2	1	10.8	271.0613	151.0045, 191.2330	C_15_H_12_O_5_

* Extract: type of extracts; 1: tomato pomace (peel, outer skins and seeds)—TP; 2: tomato peel and skins—TPS; 3: the tomato protein extract—TPE.

**Table 2 molecules-26-06403-t002:** Protein extraction from wet tomato seeds using consecutive high shear mixing extractions.

	Dry Weight (g)	Protein Concentration (mg Per g Dry Material)	Protein Yield (%)
Hulls	113.4	58.1 ^a^	12.0
First extract	42.8	245.2 ^b^	19.2
Second extract	18.6	190.7 ^b^	6.5
Third extract	7.4	199.3 ^b^	2.7
Total extracts	68.9	-	28.3
Extracts and Hulls	182.3	-	40.4
Seeds	200.0	274 ^c^	100.0

^a^ Hulls; ^b^ crude protein from each extraction; ^c^ seeds and protein concentration determined by Kjeldahl analysis.

## Data Availability

Not applicable.

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
