# Peer review of "HPLC-DAD-qTOF Compositional Analysis of the Phenolic Compounds Present in Crude Tomato Protein Extracts Derived from Food Processing"

_molecules, 2021, doi:10.3390/molecules26216403_

Round 1

Reviewer 1 Report

The manuscript is focusing on identification of phenolic acids and flavonoids in different tomato processing fractions including tomato protein extract. The introduction is well written and comprehensive. The samples preparation, the method used for analysis and identification of compounds are correctly described. After reviewing, my comments concerning the manuscript are general positive and it could be accepted in present form.

Author Response

Dear Reviewer,

thank you for your comments.

With kind regards,

Your sincerely,

Dr. Ana Miklavčič Višnjevec

Reviewer 2 Report

The manuscript highlights that preparation of food products creates residues and by-products dangerous to the environment.

Given that they still contain bioactive compounds, their reuse can have dual positive function: respect for the environment by reusing residues and replacement of synthetic additives (added in several food processing) with natural antioxidants.

Particularly, this study focused on protein extraction and identification of phenolic compounds from tomato pomace, peel and skins.

In my opinion, the following points should be revised before publishing.

  • Add the retention time information in table 1, as well as the peak number of each compound must to be inserted in the text, otherwise the reading is very complicated. For instance, please write: kampferol-3-O-rutinoside (31) was identified …. (line 189).
  • In table 1, the rt of eriodictyol-O-glucoside is not 8.3.
  • Where are the fragments at m/z 308 and m/z 285 in table 1?
  • Parent ion is not a fragment ion, therefore it should be removed from the fragments column.
  • Please, add the accurate mass of fragments to improve their confirmation.
  • The quantification method displayed in figure 1 is not clear. Did you perform data normalization?
  • Given the dependence of amount of phenolic compounds on various factors, an absolute quantification (at least of the main compounds) should be performed allowing useful comparison among different cultivars, ripeness degree, harvest time, …. See J Food Sci Technol (September 2018) 55(9):3453–3461.
  • Please check the spelling of the phrases:

Line 83. Lines 234-235-236.

Author Response

Dear Reviewer,

We are grateful for your letter dated 8th October 2021. We found your suggestions and comments very useful and helpful for improving the quality of our manuscript. The revised version of the manuscript is enclosed.

The responses to the comments and suggestions are highlighted below.

The manuscript highlights that preparation of food products creates residues and by-products dangerous to the environment.

Given that they still contain bioactive compounds, their reuse can have dual positive function: respect for the environment by reusing residues and replacement of synthetic additives (added in several food processing) with natural antioxidants.

Particularly, this study focused on protein extraction and identification of phenolic compounds from tomato pomace, peel and skins.

In my opinion, the following points should be revised before publishing.

Add the retention time information in table 1, as well as the peak number of each compound must to be inserted in the text, otherwise the reading is very complicated. For instance, please write: kampferol-3-O-rutinoside (31) was identified …. (line 189).

The retention times were added in the Table 1. The peak number of each identified compounds was written as it was suggested.

In table 1, the rt of eriodictyol-O-glucoside is not 8.3.

“In the protein extract, eriodictyol-O-glucoside (32) was tentatively identified at retention time 8.3, with a corresponding fragmentation pattern as previously reported by Vallverdu-Queralt et al. [25]. Both the deprotonated molecule (m/z 449) and hexoside moiety (m/z 287) possessed similar m/z of 449 at a retention time of 8.3 (32).”

was replaced with

“In the protein extract, eriodictyol-O-glucoside (32) was tentatively identified at retention time 8.4, with a corresponding fragmentation pattern as previously reported by Vallverdu-Queralt et al. [25]. Both the deprotonated molecule (m/z 449) and hexoside moiety (m/z 287) possessed similar m/z of 449 at a retention time of 8.4 (32).”

Where are the fragments at m/z 308 and m/z 285 in table 1?

“A bitter-tasting flavonol glycoside, kampferol-3-O-rutinoside (31) was identified in the TP and TPE, with a retention time of 8.5, and based on accurate mass determination and a typical fragmentation pattern, with a deprotonated ion (m/z 593) and the loss of a rutinoside unit (m/z 308) with m/z 285 (Table 1).” was replaced with

“A bitter-tasting flavonol glycoside, kampferol-3-O-rutinoside (31) was identified in the TP and TPE, with a retention time of 8.5, and based on accurate mass determination and a typical fragmentation pattern, with a deprotonated ion (m/z 593) and the loss of a rutinoside unit (308 Da) with m/z 285 (Table 1, peak number 31)

Parent ion is not a fragment ion, therefore it should be removed from the fragments column.

In MS/MS it is possible to detect a fragment ion with the same m/z as the parent ion. For example, the main fragments of kamepferol-3-O-rutinoside were 593 and 287 as previously reported by Vallverdú-Queralt et al., 2011.

Please, add the accurate mass of fragments to improve their confirmation.

The accurate mass of fragments have been added (Table1).

The quantification method displayed in figure 1 is not clear. Did you perform data normalization?

“Heatmap for semi-quantitative comparison of individual phenolic compound identified in the tomato pomace containing seeds, tomato protein and tomato skins and peels. Calculations were performed based on areas of MS extracted ion chromatogram (EICs), corrected for dilution during sample preparation.” was replaced with

“Heatmap for semi-quantitative comparison of individual phenolic compound identified in the tomato pomace containing seeds, tomato protein and tomato skins and peels. Calculations were performed based on areas of MS extracted ion chromatogram (EICs). The data normalization was performed including the correction for dilution during sample preparation.”

Given the dependence of amount of phenolic compounds on various factors, an absolute quantification (at least of the main compounds) should be performed allowing useful comparison among different cultivars, ripeness degree, harvest time, …. See J Food Sci Technol (September 2018) 55(9):3453–3461.

Line 363: “In addition, the main identified phenolic compounds were quantified: caffeic acid-O-hexoside (3), homovanillic acid glucoside (4), chlorogenic acid (5), caffeic acid-O-hexoside 2 (6), rutin-O-hexoside (7), cryptochlorogenic acid (8), caffeic acid (11), ferulic acid glucoside 1 (12), vicenin-2 (13), protocatechuic acid (14), coumaric acid 2 (18), rutin-O-pentoside (19), rutin (23), naringenin-O-glucoside 3 (34), eriodictyol (36), quercetin (37) and naringenin 1 (38). Phenolic compounds were quantified using the response factor for chlorogenic acid (Sigma-Aldrich, Merck KGaA, Darmstadt, Germany). The calibration plots indicated good correlations between peak areas and commercial standard concentrations. LOQ was determined as the signal-to-noise ratio of 10:1 and amount to 0.1 mg/kg dry weight sample. The standard deviation between duplicates was less than 7%.” was added.

Line 239:

“TPE contains 220 ± 15 µg/g dry weight of total determined phenolic compounds. This concentration is in the same range as the levels of total phenolic compounds determined by HPLC-DAD-MS/MS in dry tomato [19].”

Please check the spelling of the phrases:

Line 83. 

“In one study the tomato peels the main phenolic compounds present are quercetin..”

was replaced with

“In one study the main phenolic compounds present in the tomato peel were identified as quercetin..”

Lines 234-235-236.

“In accordance with our observations the final conclusion of their study was that the protein extracts evaluated in their study could be suitable to use as food additives to enhance nutritional and health related values [9].”

was replaced with

“The final conclusion of their study was that the protein extracts evaluated could be suitable to use as food additives to enhance nutritional and health related properties [9]. This is in accordance with our observations.”

We again thank you for the constructive and helpful comments. We hope that the revised manuscript will now be considered acceptable for publication.

With kind regards and best wishes,

Your sincerely,

Dr. Ana Miklavčič Višnjevec

Reviewer 3 Report

The article describes the determination of phenolic compounds in various tomato byproducts left over after tomato processing by HPLC-DAD-QTOF. The article is well written and contains few typos. I have few comments:

l. 56: Please explain the abbreviations BHA, BHT and TBHQ.

Table 1: sometimes there are dots after the numbers of isomers and sometimes there are Roman numerals and sometimes Arabic numerals - please unify them.

l.331: the phrase is abundant; it is already mentioned in the brackets in line 329

Author Response

Dear Reviewer,

We are grateful for your letter dated 8th October 2021. We found your suggestions and comments very useful and helpful for improving the quality of our manuscript. The revised version of the manuscript is enclosed.

The responses to the comments and suggestions are highlighted below.

56: Please explain the abbreviations BHA, BHT and TBHQ.

“The proposed maximum limits for synthetic phenolic antioxidants such as BHA, BHT and TBHQ indicate that the Acceptable Daily intake might be approached or exceeded in certain countries [11].”

was replaced with

“The proposed maximum limits for synthetic phenolic antioxidants such as butylated hydroxyanisole (BHA), butylated hydroxytoluene (BHT) and tertiary butylhydroquinone (TBHQ) indicate that the Acceptable Daily intake might be approached or exceeded in certain countries [11].”

Table 1: sometimes there are dots after the numbers of isomers and sometimes there are Roman numerals and sometimes Arabic numerals - please unify them.

In the Table 1 the numbers were unified.

l.331: the phrase is abundant; it is already mentioned in the brackets in line 329

“The separation was performed at a flow rate 0.5 mL/min, using a 1µL injection volume.”

was omitted.

We again thank you for the constructive and helpful comments. We hope that the revised manuscript will now be considered acceptable for publication.

With kind regards and best wishes,

Your sincerely,

Dr. Ana Miklavčič Višnjevec

Round 2

Reviewer 2 Report

the table in Figure 1 is not still clear.

what is the meaning of numbers from 5 to 235?